# Screening for Autism Spectrum Disorder in Premature Subjects Hospitalized in a Neonatal Intensive Care Unit

**DOI:** 10.3390/ijerph17207675

**Published:** 2020-10-21

**Authors:** Norrara Scarlytt de Oliveira Holanda, Lidiane Delgado Oliveira da Costa, Sabrinne Suelen Santos Sampaio, Gentil Gomes da Fonseca Filho, Ruth Batista Bezerra, Ingrid Guerra Azevedo, Silvana Alves Pereira

**Affiliations:** 1Physiotherapy Course, Universidade Federal do Rio Grande do Norte (UFRN), Natal 59078-970, Rio Grande do Norte, Brazil; scarlyttnorrara23@gmail.com (N.S.d.O.H.); lididelgado@gmail.com (L.D.O.d.C.); apsilvana@gmail.com (S.A.P.); 2Post-graduation Program of Physiotherapy, Universidade Federal do Rio Grande do Norte (UFRN), Natal 59078-970, Rio Grande do Norte, Brazil; sabrinne.suelen@gmail.com (S.S.S.S.); gentilfonsecafisio@gmail.com (G.G.d.F.F.); 3Instituto Santos Dumont, Macaíba 59280-000, Rio Grande do Norte, Brazil; 4Rehabilitation Sciences Graduate Program, Faculty of Health Sciences of Trairi/ Universidade Federal do Rio Grande do Norte (FACISA/UFRN), Santa Cruz 59200-000, Rio Grande do Norte, Brazil; ruthbezerrafisio@gmail.com; 5Department of Therapeutic Processes, Universidad Católica de Temuco, Temuco 4813302, La Araucania, Chile

**Keywords:** autism spectrum disorder, risk factors, newborn, neonatal intensive care units

## Abstract

Considering that the average age for diagnosis of autism spectrum disorder (ASD) is 4–5 years, testing screening methods for ASD risk in early infancy is a public health priority. This study aims to identify the risks for development of ASD in children born prematurely and hospitalized in a neonatal intensive care unit (NICU) and explore the association with pre-, peri- and postnatal factors. *Methods*: The children’s families were contacted by telephone when their child was between 18 and 24 months of age, to apply the Modified Checklist for Autism in Toddlers (M-CHAT). The sample consisted of 40 children (57.5% boys). M-CHAT screening revealed that 50% of the sample showed early signs of ASD. Although the frequency of delayed development was higher in boys, this difference was not statistically significant between the sexes (*p* = 0.11). Assessment of the association between perinatal conditions and early signs of autism in children hospitalized in an NICU exhibited no correlation between the factors analyzed (birth weight and type of delivery). The findings indicate a high risk of ASD in premature children, demonstrating no associations with gestational and neonatal variables or the hospitalization conditions of the NICUs investigated.

## 1. Introduction

During its intrauterine life, the fetus’ systems and subsystems, indispensable for extrauterine life, are in constant structural and chemical maturation [1]. Premature birth interrupts this structural development, which then becomes dependent on the care of a neonatal intensive care unit (NICU) [1,2]. Despite the fact that the NICU provides essential life support for the newborn, internal organic factors and intense exposures may affect the trajectory of its development [3,4]. The longer the NICU stay, the greater the benefit and recovery of neuromotor and cognitive functions [5].

Exposure to numerous clinical procedures associated with noise [6], intense lighting [7] and the use of devices characteristic of the NICU [8] induces stress and/or pain in the newborn, triggering functional [9], microstructural [10] and metabolic [11] alterations in widely disseminated regions of the brain [11]. As a clinical alternative, analgesic and sedative practices aim at minimizing pain and stress related to clinical procedures. However, these agents may also be associated with compromised brain maturation [11,12], raising the crucial question of whether early administration of anesthetics and sedatives in premature newborns (PNB) is safe for the developing central nervous system.

Recent years have demonstrated the relation between NICU experiences and changes in psychic development [13]. In a systematic review, Gardener et al. [14] sought to explain the association between perinatal and neonatal factors and the risk of autism spectrum disorder (ASD) [14], concluding that there is no exact description of how these events relate to the risk of ASD. Although studies that developed and tested streamlined methods for ASD diagnosis provide evidence on exposure to multidimensional factors in the neonatal period, the average age of screening or diagnosis is 4 to 5 years [15,16,17].

Limited access to timely ASD screening evaluations delay enrollment in interventions known to improve developmental outcomes [15]. Considering the hypothesis that identifying the risks for the development of ASD in early infancy may help in early diagnosis and that this is a public health priority, the present study aimed to identify the risks for development of ASD in children born prematurely and hospitalized in an NICU and explore the possible association with pre-, peri- and postnatal factors.

## 2. Materials and Methods

This is a prospective longitudinal study of children born prematurely and hospitalized in 2017 in the NICU of a maternity school specialized in high-risk births. It was approved by the Research Ethics Committee, under protocol number 1.707.627/2016, in line with National Health Council Resolution 466/12. All the parents or legal guardians gave written informed consent, authorizing the use of the children’s data in the research.

The newborns were recruited using non-probability sampling and selected by convenience. The eligibility criteria considered the medical records of all PNB (gestational age < 37 weeks) admitted to the NICU between 1 January and 30 November 2017, and records with incomplete data or reporting genetic, neurologic or metabolic alterations were excluded as confounders in the data analysis. The gestational and neonatal variables were collected from virtual medical records on the University Hospitals Management App. (AGHU, Ministério da Educação, Brasília, Brazil) and in the NICU daily monitoring documents.

In order to assess the risk of the PNBs having developed a social disorder in early childhood, the baby’s family was contacted by telephone when their child was between 18 and 24 months of age. When telephone contact was unsuccessful, an active search of social media was conducted to schedule the call. The mothers completed the Modified Checklist for Autism in Toddlers (M-CHAT) and when doubts arose, were asked to visit the outpatient care unit for further clarification.

The M-CHAT, which consists of 23 “yes” or “no” questions, was applied according to the scoring criteria, considering items 2, 7, 9, 13, 14 and 15 as critical for the risk of ASD [18]. After the instrument was scored and the risk criteria determined, the children were divided into two groups: “risk” and “no risk”, in line with the M-CHAT application manual, which defines “risk” as a child who scores on at least two of the six critical items contained in the instrument [19].

The Statistical Package for the Social Sciences, version 20.0 (SPSS, IBM, Armonk, NY, USA), was used in the statistical analysis, with data expressed as measures of central tendency and dispersion in a frequency table to characterize the sample. The Kolmogorov–Smirnov test was conducted to assess data normality. The Student’s *t*-test for independent samples or the Mann–Whitney test were applied to analyze the quantitative variables, depending on data normality, and the chi-squared test for associations between the categorical variables and the presence of risk, as determined by the instrument. Significance was set at *p* < 0.05. The sample size was based on the 2019 Maia study [20]. The power of the study was established at 80%; significance was set at 0.05 and a 0.18 probability of exposure among the controls. The sample size required was 36 subjects.

## 3. Results

During the study period, 372 newborns were admitted to the NICU, 149 of whom met the inclusion criteria. Of these, 109 were excluded due to failed telephone scheduling (72) or incomplete medical records (37), and 40 families that completed the ASD screening test were included in the study. The descriptive analysis of the 40 PNBs assessed between 18 and 24 months of corrected age (31.25 ± 2.90 weeks) is presented in Table 1.

With respect to the risk of developing ASD, 20 children exhibited early risk for ASD, 70% of whom were boys. However, this difference was not statistically significant between the sexes (*p* = 0.11). Bivariate analysis found no significant associations between the categorical (Table 2) or quantitative variables (Table 3) analyzed and ASD.

## 4. Discussion

The data of this study demonstrate that 50% of children born prematurely and hospitalized in the NICU, assessed after the age of 1 year, exhibited early risk for ASD. However, no associations were found between the risk of ASD, gestational and neonatal variables and hospitalization conditions. Gadassi et al. [21] also applied M-CHAT in premature newborns hospitalized in an NICU; however, of the 110 premature babies evaluated, only 33 displayed risk for ASD. The difference between the percentage at risk for ASD between the two studies is likely due to a combination of pre-, peri- and postnatal periods [11,22].

Wang et al. [23] reported that low birth weight (risk ratio—RR = 1.26; confidence intervals 95%—CI: 1.20, 1.34; *p* < 0.001), postpartum hemorrhage (RR = 2.10, 95% CI: (1.30, 3.40); *p* = 0.002), male sex (RR = 1.47, 95% CI: (1.39, 1.55); *p* < 0.001) and brain anomalies (RR = 5.38, 95% CI: (1.16, 24.9); *p* = 0.031) are risk factors for ASD. One hypothesis to justify this association is that the combination of maternal, neonatal and postnatal risk factors, which culminates in longer neonatal hospitalization, increases the risk of outcomes harmful to neurodevelopment and brain maturation. Several authors corroborated this hypothesis and found that the combination of these factors increases the risk for ASD [8,11,19,22,24,25,26,27,28].

In the present study, we believe that aspects such as homogeneity, a small sample size and the limited number of factors analyzed contributed to the lack of association between the conditions investigated. This may be due to the broader hypothesis that events in the intrauterine, neonatal and childhood environment may interact and/or contribute to ASD, in combination with other cofactors (environment and genetic dimensions) [28].

Dudova et al. [29] screened for the risk of ASD in premature babies using a battery of tests that included the M-CHAT, the Communication and Symbolic Behavior Scales-Developmental Profile-Infant Toddler Checklist (CSBS-DP-ITC) and the Infant/Toddler Sensory Profile (ITSP). In their results, 42.7% of the children evaluated showed a positive result in at least one of the screening questionnaires, a similar finding to that of our study, since it demonstrated a high risk of ASD in the children investigated. However, the authors also did not associate the risk of ASD to the combined factors, observing only that prematurity seems to be a nonspecific risk factor when compared to other known factors, such as tuberous sclerosis and genetic syndromes.

Wong et al. [30] used the quantitative checklist for autism in children (Q-CHAT) to characterize early childhood social communication skills and autistic traits in children born very prematurely [30]. When comparing the Q-CHAT scores with the published scores of the general population, they found that with an average chronological age of 24 months, the Q-CHAT scores of the preterm cohort (mean of 33.7, SD 8.3) were significantly higher than those of the general population (mean 26.7; SD 7.8), indicating greater difficulty in social communication and autistic behavior [30]. These outcomes corroborate the results found in our study when considering the critical items scored by the children evaluated, revealing that screening in children aged 18 and 24 months can assist in early detection, and are consistent with the current recommendations of the American Academy of Pediatrics [31].

In order to examine the performance of the First Year Inventory (FYI; version 2.0), a group of American researchers applied the screening instrument to a high-risk sample of 12-month-old children with older siblings diagnosed with ASD [32]. Babies subsequently diagnosed with ASD had higher FYI 2.0 risk scores in the social communication and sensory regulation domains when compared to those with typical development [32]. Although the present study is limited for not conducting a diagnostic follow-up, these data corroborate the main complaints reported and identified on the M-CHAT, revealing prodromal signs of ASD.

With a view to removing cultural barriers still present in the use of M-CHAT in some areas of the world [33], Perera et al. [34] investigated a new tool for ASD diagnosis: the pictorial autism assessment schedule (PAAS). This 21-question instrument is accompanied by photos illustrating each question and showed a sensitivity of 88.8% in the differentiation between ASD and other developmental disorders, and 80% in the distinction between ASD and typical development. In a previous study with M-CHAT [33], the instrument obtained a sensitivity of 25%. Parera et al. [34] underscore the importance of cultural adaptation when applying a tool in the population and show that no instrument is universally effective for all samples, making it important for researchers to be aware of possible flaws in the tool.

In a case–control study based on the live birth records of a reference hospital in Finland, Polo-Kantola et al. [24] proposed that obstetric risk factors and the risk for ASD be examined. The results showed that ASD was associated with maternal high blood pressure (OR 1.49, 95% CI: (1.1, 2.1), *p* = 0.018), Apgar indices less than 7 (1 min, OR 1.46, 95% CI: (1.1, 2.0), *p* = 0.021) and neonatal treatment with monitoring in an NICU (OR 1.40, 95% CI: (1.02, 1.9), *p* = 0.038). According to the authors, these discoveries reveal that low Apgar indices in the first minute may indicate fetal suffering, raising the hypothesis that acute or prolonged fetal oxygen deprivation may be a risk factor for various neuropsychiatric disorders, including ASD.

Although we found no association between ASD and pre-, peri- and postnatal factors, the risk frequency for ASD in PNBs hospitalized in an NICU is high and an important aspect to investigate [24]. Winkler-Schwartz and Garfinkle underscore the need for early screening, preferentially in newborns monitored in an NICU [25], especially when the prevalence of ASD in school-age children differs from that of premature children [29].

Although 149 PNBs initially met the study inclusion criteria, and some were not included in the sample, our sample size was greater than the calculated sampling. Additionally, the sample size showed a power of 0.80. The high sample loss was due to the impossibility of previous telephone contact (72 families) or incomplete medical records (37 medical records), precluding M-CHAT application and suggesting possible longitudinal monitoring. Thus, from the study sample, we observed that the frequency of ASD risk signs was higher in male children (70%). It is estimated that ASD prevalence is between 0.1 and 2% among different populations, with a greater predominance in males than females [27]. The reasons for these findings are still a matter of debate in the clinical and scientific community; however, these aspects require greater caution on the part of parents and health professionals when associating them with neurobehavioral signs typical of ASD, as well as broader diagnostic criteria.

Although some studies demonstrate that exposure to a range of environmental conditions in the NICU may compromise neuropsychomotor development [3,5,11], methodological variations culminate in heterogeneous effects of risk factors related to ASD. However, identifying the early signs of ASD and possible signs of delayed neuropsychomotor development stimulates strategies to understand the variables involved in these conditions, contributing to the adoption of preventive actions and early intervention [35,36].

The results of this study demonstrate that 50% of children who were assessed by M-CHAT after 1 year of age showed positive risk signs for ASD. It is important to underscore that diagnostic assessments were not carried out in this sample. Thus, the specificity of screening in premature infants remains uncertain, raising questions as to whether a positive M-CHAT prediction reflects autistic traits or is indicative of impaired social and communicative skills associated with preterm phenotype. Thus, we consider it important to screen and monitor premature children who were hospitalized in the NICU, in order to make health professionals and families aware of possible future health problems later in life. As such, screening the factors associated with ASD early in life would be an important tool, signaling the need to support these children and orienting their families about the possible repercussions throughout childhood, thereby avoiding future public health problems. In addition, we provide important information and inquiries that show the need for exploratory analysis of neonatal, gestational and hospitalization conditions, conferring the relevance of a longitudinal monitoring of this population born prematurely.

Despite the fact that the present study provides new evidence for the area of ASD, a number of limitations should be considered. Although the study presented a screening scale for the risk of ASD, it did not include a final diagnosis of the disease. However, we emphasize that early screening also means early diagnosis, thereby enhancing the health of a Brazilian region where intermittent outpatient follow-up is high and the number of specialized professionals is low. Another limitation is that the study assesses only gestational and neonatal conditions, which are insufficient to confirm a direct relationship with ASD, given the multifactorial characteristics of autism.

## 5. Conclusions

Although the study identified a high risk of ASD in the children evaluated, it was not possible to determine the risk factors associated with the gestational and neonatal variables and the hospitalization conditions analyzed. The children at greatest risk were predominantly male, but this variable should be further analyzed in conjunction with the others, given their etiological variability, in order to better understand their effects on ASD. Moreover, this study reaffirms the importance of applying this type of tool to ASD screening in children born prematurely.

## Figures and Tables

**Table 1 ijerph-17-07675-t001:** Sample characterization.

Variables	Total Sample (*n* = 40)
Sex	
Female	17 (42.5%)
Male	23 (57.5%)
Birth weight (grams) †	1420 (1022–2081)
Gestational age (weeks) *	31.18 (3.00)
1-min Apgar †	8 (4.25–8)
5-min Apgar †	9 (8–9)
Days hospitalized *	35.63 (26.43)

Categorical variable presented as *n* (%). * variable expressed as mean ± standard deviation. † variable expressed as median and 25th–75th percentile.

**Table 2 ijerph-17-07675-t002:** Bivariate analysis for the categorical variables according to the risk of developing autism.

Categorical Variables	Risk—*n* (%)	No Risk—*n* (%)	*p*-Value
**Sex**			
Female	6 (30%)	11 (55%)	0.11
Male	14 (70%)	9 (45%)	
**Type of delivery**			
Normal	8 (40%)	10 (50%)	0.52
Cesarean	12 (60%)	10 (50%)	
**IH**			
Yes	7 (35%)	7 (35%)	0.92
No	8 (40%)	7 (35%)	
No diagnosis	5 (25%)	6 (30%)	
**Postnatal corticosteroid**			
Yes	5 (25%)	3 (15%)	0.69
No	15 (75%)	17 (85%)	

IH: intracranial hypertension.

**Table 3 ijerph-17-07675-t003:** Bivariate analysis of quantitative variables according to the risk of developing autism.

Quantitative Variables	Risk (*n* = 20)	No Risk (*n* = 20)	*p*-Value
Birth weight (grams) †	1278.15 (993.50–2031.20)	1764.00 (1128.50–2156.00)	0.29
Gestational age (weeks) *	31.10 ± 2.67	31.25 ± 3.37	0.31
1-min Apgar †	8.00 (4.25–8.75)	8.00 (4.50–8.00)	0.75
5-min Apgar †	9.00 (8.00–9.00)	9.00 (8.00–9.00)	0.88
Days on MV †	3.00 (1.25–30.75)	8.50 (00–15.00)	0.18
Days on oxygen therapy †	10.00 (3.25–41.50)	5.50 (2.00–31.25)	0.30
Days hospitalized *	36.15 (22.00)	35.10 ± 30.87	0.51

MV: invasive and noninvasive mechanical ventilation. * variable expressed as mean ± standard deviation. † variable expressed as median (25th–75th percentile).

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
