# Peer review of "Screening for Autism Spectrum Disorder in Premature Subjects Hospitalized in a Neonatal Intensive Care Unit"

_ijerph, 2020, doi:10.3390/ijerph17207675_

Round 1

Reviewer 1 Report

In their research article, the Authors aimed to identify the risks for development ASD in children that had born prematurely and hospitalized in a NICU and to explore the association with pre, peri and postnatal factors. They contacted the children’s family by telephone when their child was between 18 and 24 months of age, to apply the Modified Checklist for Autism in Toddlers. Among 40 children (57.5% boys) they found that 50% of the sample showed early signs of ASD and a high risk of ASD in the premature children, demonstrating no associations with gestational and neonatal variables or the hospitalization conditions of the investigated NICUs.

The paper is of interest for the research question, mainly in a peculiar geographic context, such as Brazilian region. Methods are appropriate and references quite adequate. Its readability is quite good. Of value, the early assessment (18-24 months) chosen. However, the small sample size limits the conclusions. Indeed, it’s lacking the sub analysis of different gestational ages at birth, able to better understand the real impact of NICU admission and stay duration on ASD risk, as well as the missed final diagnosis of the disease. In the opinion of this reviewer, the paper could be of effective interest after an implementation of sample size and appropriate statistical analysis, without generating premature fears in parents and alarms in clinicians.

Author Response

We appreciate the reviewers comments and following we send the answers to each request.

Reviewer 2 Report

Following are the major suggestions/comments that authors may address to improve the manuscript for the scientific community and readers of Children: 

  1. Compare and contrast various ASD screening instruments in the Discussion.
  2. Please report and discuss the number records were with incomplete data/reporting.
  3. Only few variables (table 2 and 3) compared for the risk for developing autism.  Can the authors provide explanation on selection of these variables while not including other variables.
  4. The authors found higher frequency of the delayed development in boys.  Please elaborate and discuss.
  5. Long-term follow-up beyond 2 years would have been valuable and would have confirm the diagnosis of ASD. 
  6. Discussion is shallow and needs further finishing.  Please discuss what are the novel observations of the paper and how they advance the current understanding of the field.

Author Response

(The authors gave the same response as above.)

Reviewer 3 Report

Thank you for asking me to review the manuscript titled, Screening for autism spectrum disorder in premature subjects hospitalized in a neonatal intensive care unit.” Below are my comments and suggestions, section-by-section.

Title

  • The title adequately describes the study.

Abstract

  • Listing an example or two of the factors analyzed (line 35) would be helpful
  • Otherwise, the abstract is clear and concise.

Introduction

  • Line 64 should be “for the development of ASD…”

Materials and Methods

  • Line 85 - please use ASD in place of “autism” here and beyond when appropriate.
  • M-CHAT risk categories should be reported (i.e., how many were in low vs moderate vs high risk). Also, it should be stated that “Risk” in Table 2 was comprised of moderate and high-risk groups (which I assume is what was done)?
    • Further, was the M-CHAT or used M-CHAT-R or M-CHAT-R/F?
  • Did you also look at the relationship between APGAR scores and later M-CHAT scores? It would be nice to see this relationship in Table 3.

Discussion & Conclusion

  • The Discussion is well written.
  • Do the authors plan on following-up with the participants a year or so later to see if an actual ASD diagnosis was made? That would be very interesting data to see.

Overall, this short report is very well-written, informative, and is an important addition to the research literature in ASD.

Author Response

(The authors gave the same response as above.)

Round 2

Reviewer 1 Report

The Authors have addressed the concerns in an acceptable way.

Author Response

We thank the reviewer for the helpful suggestion provided in the round 1 of the review process.